# Influence of Crystalline Admixtures and Their Synergetic Combinations with Other Constituents on Autonomous Healing in Cracked Concrete—A Review

**DOI:** 10.3390/ma15020440

**Published:** 2022-01-07

**Authors:** Yuanzhu Zhang, Runwei Wang, Zhi Ding

**Affiliations:** 1Department of Civil Engineering, Zhejiang University City College, Hangzhou 310015, China; zhangyz@zucc.edu.cn (Y.Z.); dingz@zucc.edu.cn (Z.D.); 2College of Civil Engineering and Architecture, Zhejiang University, Hangzhou 310058, China

**Keywords:** crystalline admixtures (CAs), cementitious composites, autonomous healing, healing products

## Abstract

Crystalline admixtures (CAs) are new materials for promoting self-healing in concrete materials to repair concrete cracks. They have been applied to tunnel, reservoir dam, road, and bridge projects. The fundamental research and development of CAs are needed concerning their practical engineering applications. This paper reviews the current research progress of commercial CAs, including self-made CA healing cracks; the composition of CA; healing reaction mechanism; the composition of healing products; distribution characteristics of healing products; the influence of service environment and crack characteristics on the healing performance of CA; and coupling healing performance of CA with fiber, expansive agent, and superabsorbent polymers. The current research findings are summarized, and future research recommendations are provided to promote the development of high-performance cement matrix composites.

## 1. Introduction

Cracking is an unavoidable phenomenon in concrete structures. In the process of concrete strength development, external or internal cracking is inevitable. The continuous expansion of cracks can cause a gradual decline in concrete durability. It makes it easier for corrosive ions to penetrate the concrete, destroy the microstructure of concrete and affect its macro performance. In recent years, adding some substances in cement-based materials to make the structure self-healing without intervention has gradually become the focus of research, because this method can significantly improve the durability and sustainability of the structure. For this reason, attempts have also been made to define proper test methodologies that can quantify the improvement in durability as a function of the specific self-healing mechanism [1,2]. Adding crystalline admixtures (CAs) is one of the self-healing techniques. Some scholars used commercial crystalline admixtures (component is unknown), and some scholars used alkali-active materials such as hydroxide, sodium silicate, carbonate, and metakaolin as crystalline admixtures [3,4,5,6,7,8,9,10].

Cement-based materials have some healing performance. The contact of moisture with the crack’s surface will facilitate the production of healing products. The crystal morphology of healing products is shown in Figure 1 [11]. Some researchers [12,13,14] used X-ray CT to scan the cracks before and after the healing of ordinary mortar specimens. It was found that the healing products mainly grow at the crack mouth, as illustrated in Figure 2. Sisomphon et al. [15] inferred that since CO_3_^2−^ and HCO_3_^−^ were rarely released from the matrix, external water was the only source of CO_3_^2−^ and HCO_3_^−^ in the fracture solution. The schematic diagram of ion concentration is shown in Figure 3. There are more Ca^2+^, CO_3_^2−^, and HCO_3_^−^ near the crack mouth, and CaCO_3_ is the most likely precipitate.

The ordinary Portland cement (OPC) hydration products such as C-S-H, Aft, and Ca(OH)_2_ (appropriate amount of dihydrate gypsum is added into cement as retarder, and AFm, C-A-H, and other products will be generated after hydration) prevent further hydration of cement particles [16]. Linderoth et al. [17] found that the hydration degree of cement mortar with a water–cement ratio of 0.50 was 92.1% after one year, and the hydration degree of cement mortar with a water–cement ratio of 0.30 was only 67.6%. Courtial et al. [18] reported that the hydration degree of high-performance concrete with a water–binder ratio of 0.16 was only 28~30% after 28 days of hydration. Tuan et al. [19] found that the hydration degree of high-performance concrete with a water–binder ratio of 0.18 was only 30~40% after 91 days of hydration. Slag cementitious materials (SCMs, such as blast furnace slag, fly ash, and metakaolin) react slower compared to cement, and thus more unreacted binder material remains available. Having a good chloride penetration resistance, SCMs are often used in coastal and marine structures [20,21,22].

The healing mechanism is as follows: (1) Partially hydrated cement particles and other cementitious materials (pulverized fly ash, blast furnace slag, and so on) on the crack surface will further hydrate, and C-S-H, Aft, and portlandite may exist in the healing products. (2) Free Ca^2+^ ions react with CO_2_ dissolved in water (HCO_3_^−^) to form calcite or aragonite (CaCO_3_) crystalline phases in water. In seawater, crystalline phases such as brucite, aragonite, calcite, ettringite, gypsum, and Friedel’s salt will be generated due to corrosive ions such as Cl^−^, Mg^2+^, and SO_4_^2−^. (3) Expansion is caused by the hydration of cementitious materials within a specific range of crack surfaces. (4) The blocking effect due to broken particles or impurities in water facilitates the cracks’ closing [23,24,25]. The main influential factors of crack healing include the following: (1) the composition of cementitious materials, water–binder ratio, and age of matrix; (2) healing environment; (3) width and shape of cracks [26].

However, the healing ability of OPC is minimal. Palin et al. [27] found that the upper limit of crack width of ordinary mortar specimens completely healed by underwater curing was 168 μm. Wu et al. [4] considered that only the cracks with the maximum width of 50~100 μm can completely heal at a slow speed. Cuenca et al. [28] reported that the maximum width of fully healed fractures was 30–50 μm. Therefore, other materials need to be incorporated to improve the healing performance of cement-based composites.

Crystalline admixture (CA), sometimes called cementitious capillary crystalline waterproofing material (CCCW), can completely heal cracks with widths up to 0.4 mm. CA has high-cost performance and convenient construction compared with shape memory alloy, bacteria, microcapsules, and other crack self-healing materials. The healing product has good compatibility with the matrix [29]. CA can also improve the overall strength and impermeability of the structure by enhancing porosity [5,30,31,32,33]. Some studies have noticed that the mechanical strength of CA specimens is 7~10% higher than that of ordinary specimens [34].

The well-known CA brands such as Xypex (Vancouver, BC, Canada) and Penetron (New York, NY, USA) have been successfully applied to the phase 2 cofferdam of the Three Gorges Project (Yichang, China), Park Avenue Tunnel (New York, NY, USA), Tassahra Hydropower Station (Medellin, Colombia), Expo Tunnel (Milan, Italy), and other projects. Various manufacturers have provided a series of crystal material products, mainly including the following: surface-applied spraying or coating, integral waterproofing admixtures, and sealing and repairing mortars [35]. Engineers can choose appropriate crystalline materials according to the specific circumstances of the building.

The fundamental research and development of CAs are challenging due to the confidentiality of CA components and the diversity of series products. In order to promote the development of cement-based composites with CAs and improve the concrete durability, this paper presents a review on the reaction mechanism of CA; composition and distribution of healing products; the influence of service environment and crack characteristics on the healing performance of CA; and synergetic healing performance of CA with fibers, expansive agent, and superabsorbent polymers (SAPs). The main focus of the paper is on the fracture healing performance of CA.

## 2. Commercial Crystallization Admixture

### 2.1. Composition

Some studies reported the use of commercial CAs directly in experiments. The use of CA in laboratory testing is shown in Table 1. Commercial CA is usually made of ordinary Portland cement, quartz sand, and other active substances. A significant number of microstructural investigations such as XRF [9,10,15,36,37,38,39], XRD [37,38,40], SEM [39,41,42], EDS [42], FTIR [40], and particle size distribution test [37,38] have been conducted to characterize the elemental composition of the commercial CAs. Major conclusions are as follows:(1)The test results of XRF shown in Table 2 show the following: (a) Compared with ordinary Portland cement, CAs contain more Na_2_O, especially Penetron. The high alkali environment formed by a high concentration of Na_2_O will significantly promote the crystallization reaction of alkali-silicate, generate precipitation, and increase the concentration of CO_2_ in the solution. Then, the hydration products will be carbonized in the form of volume expansion. (b) The results of CA detection are rather different, even for the same brand of CA, especially the MgO content. The highly alkaline environment created by Na_2_O increases the expansion of Mg(OH)_2_ crystal [39,43].(2)Guzlena et al. [40] investigated five brands of CA (specific brands were not mentioned). The FTIR and XRD patterns of CA-A and CA-D were similar, but the healing effect differed.

Therefore, it is difficult to obtain valuable results from these tests.

**Table 1 materials-15-00440-t001:** Application of commercial CAs and self-made CAs in laboratory test.

CA	Recommended Dosage from Manufacturer (wt%)	Dosage in Laboratory Test (by Weight of Cementitious Materials, wt%)	References
Xypex	0.8~2.0 (concrete) 2.0~3.0 (mortar)	1.5~6	[9,10,15,36,38]
Penetron	1~1.5	0.5~1.5	[28,39,41,42,44,45,46,47]
Kryton	2	2	[31]
Sika	2	2	[48]
Not mentioned	-	-	[5,40,49,50,51,52,53]
Self-made	-	-	[4,6,7,29,54,55,56]

**Table 2 materials-15-00440-t002:** Chemical composition of crystalline materials from XRF test(wt%).

	CaO	SiO_2_	Al_2_O_3_	Fe_2_O_3_	MgO	SO_3_	Na_2_O	K_2_O	Others	References
Xypex	73.40	13.72	3.66	2.28	0.70	3.91	1.24	0.40	0.69	[15,36]
Xypex	59.77	8.10	1.98	2.08	0.82	2.09	1.29	0.44	<0.2	[37]
Xypex	53	16	3.99	4.05	15.9	3.48	2.24	0.407	0.933	[38]
Xypex	53.53	14.10	4.36	1.84	11.3	2.79	-	-	12.08	[9,10]
Penetron	47.26	13.48	3.70	1.44	3.54	2.05	11.02	0.74	16.77	[39]
Ordinary Portland cement	63.3	19.5	5.6	2.3	1.1	2.7	0.3	0.9	4.3	[15]

### 2.2. Healing Mechanism

#### 2.2.1. Mechanism of Precipitation Reaction

It was reported by American Concrete Institute (ACI) that the active substance of CA (M_x_R_x_) reacts mainly with tricalcium silicate (C_3_S) to form dense C-S-H and M_x_CaR_x_-(H_2_O)_x_ crystals (Equation (1)) [3,26,34,57].
3CaO·SiO_2_ + M_x_R_x_ + H_2_O → Ca_x_Si_x_O_x_R·xH_2_O + M_x_CaR_x_·xH_2_OCalcium silicate + crystalline promoter + water → modified calcium silicate hydrate + pore-blocking precipitate(1)

#### 2.2.2. Complexation–Precipitation Reaction Mechanism

Under the action of concentration gradient, active substances move with water. When the active material enters the high concentration area of Ca(OH)_2_, it reacts with free Ca^2+^ to form a calcium-complex precipitate that is soluble in water and unstable and diffuses with water in the structural pores and adsorbs in the capillary wall. When the calcium complex encounters incomplete hydration cementitious materials, Ca^2+^ will be seized, and the active substances will be replaced by more stable SiO_3_^2−^, AlO_2_^−^, and CO_3_^2−^ groups, resulting in complexation–precipitation reaction to generate more stable crystal compounds (hydrated calcium silicate, hydrated calcium aluminosilicate hydrate, and other crystal precipitates). When the active substance is replaced, it becomes free radical again and continues to react. Although SiO_3_^2−^ can be directly reacted with Ca^2+^, the reaction rate to produce precipitate is relatively slow at low concentrations. The particle size of the calcium complex is very small (10^−10^ m). It can react with Ca^2+^ at a low concentration (0.01 × 10^−6^), making the combination of Ca^2+^ and SiO_3_^2−^ occur easily. Moreover, calcium complexes can quickly fill the capillary pores, hinder continuous water leakage, and reduce the loss of Ca^2+^ [43,58,59].

### 2.3. Crystallization Products

The curing environment has a significant influence on the composition of healing products. The healing maintenance environment can be divided into corrosive and noncorrosive according to the concentration of corrosive ions. Table 3 lists the healing conditions in laboratory experiments. The healing products of specimens mixed with CA and the expansive agent are detailed in Section 5.2.1.

#### 2.3.1. Noncorrosive Healing Environment

In the noncorrosive healing environment, the healing products of CA specimens are mainly CaCO_3_, C-S-H, portlandite, and ettringite, but nonconsistent test results are found in different studies. Through the SEM-EDS test, Cuenca et al. [28] found that there were more cubes and needle-like products in the healing products of the CA specimen but no such products in the reference specimen. Through the SEM-EDS test, Ferrara et al. [42] found that there were more acicular products on the crack surface of the healed CA specimen, while there was no such acicular product on the crack surface that has just cracked. Li et al. [46] found that the healing products of the CA and reference specimens were mainly portlandite and calcite by XRD test. The CA specimen has more crystalline products than the reference specimen, and portlandite has a higher XRD peak. Through the SEM test, Escoffres et al. [48] found that the healing products of reference specimens were mainly calcite and ettringite, and the healing products of CA specimens were mainly aragonite (CaCO_3_). Through XRD and TGA test, Xue et al. [39] inferred that CA promoted the growth of AFt to promote fracture healing, while the healing products of the reference specimens were mostly CaCO_3_ crystals.

Huang et al. [25] reviewed the self-healing products of ordinary Portland cement materials and found inconsistent healing products in different studies, revealing that the admixture of CA makes this reaction more complex. It is speculated that the inconsistency of test time, crack width, and cementitious materials led to different healing products [11,60,61]. The wider the crack width is, the longer the healing time is, and the more easily the calcium hydroxide is carbonized into calcium carbonate precipitation. More C-S-H will also carbonize.

As per recent studies [6,8,28,36,38,41,42,45,48,62,63], the needle-like healing products were characterized as ettringite crystals (AFt, 3CaO·Al_2_O_3_·3CaSO_4_·32H_2_O) through microstructural investigations. However, some microstructural studies reported non-ettringite needle-like products. Liu [43] observed needle-like and fibrous crystals with different growth patterns from ettringite. SEM/EDS analysis was used to characterize the crystal elements. It was inferred that the needle-like product was a hydrated sodium silicate crystal, and the fibrous crystal contained Na and Mg elements which do not exist in ettringite. Azarsa et al. [64] compared the crystalline phases of healing products of three brands of CA by SEM-EDS, and needle-like products were observed in the three specimens. The EDS analysis of needle-like products of two samples was similar to that of ettringite, but the EDS analysis of one sample did not detect the peak of the S element (Figure 4). Therefore, the authors inferred that the needle-like product was not ettringite.

In the noncorrosive curing environment, calcite (CaCO_3_) is the main healing product of CA and reference specimens. CaCO_3_ can exist in various forms depending on the healing environment [48]. Calcite and aragonite (Figure 5) are the primary crystalline forms of CaCO_3_ in the healing products as per the existing studies. Escoffres et al. [48] found that Mg in CA promoted the formation of aragonite. It can be inferred from the seepage–tensile stress coupling test results that the healing of the CA specimen is slower than that of the reference specimen. However, the mechanical properties of the CA specimen recover better. The authors found that the possible reason is that the growth rate, bonding degree, and compactness of aragonite and calcite are different. Wang et al. [9] analyzed the healing products of specimens mixed with CA, Na_2_CO_3_, CSA, and CaHPO_4_ by SEM-EDS. Cluster-like CaCO_3_ crystals were observed in the cracks in addition to diamond-like CaCO_3_ crystals. The incorporation of Na_2_CO_3_ provides more CO_3_^2−^ for the solution in the fracture, so the authors concluded that the concentration of CO_3_^2−^ can affect the morphology of CaCO_3_ crystals.

#### 2.3.2. Corrosive Environment

Existing studies [65,66,67,68,69] suggest that a corrosive environment could significantly reduce the durability of cracked concrete. However, corrosive ions can also react with cement to generate crystals that can partially heal cracks.

Seawater contains high concentrations of Cl^−^, SO_4_^2−^, Mg^2+^, and Na^+^, which can react with hardened cement to produce brucite, ettringite, CaCO_3_ (aragonite or calcite), magnesium silicate hydrate, Friedel’s salt, gypsum, CaCl_2_, Ca(HCO_3_)_2_, etc. [27] Liu et al. [70] found that the crack closure rate of seawater curing specimens is 2.5 times that of water curing specimens. Seawater significantly promotes the healing of cracks. TGA test showed that Mg^2+^ was the primary reaction ion in the seawater curing environment, rather than the Cl^−^ with the most content. Brucite and CaCO_3_ are the main healing products, and almost no chlorine exists in the healing products. Palin et al. [27,71] found that the healing products of cracked specimens under seawater curing were mainly brucite and aragonite. A brucite layer was rapidly formed on the fracture surface within two days, and then aragonite layer precipitation was slowly formed on the brucite layer. The test results reported that the maximum crack widths of the OPC mortar specimens cured in seawater and tap water are 0.592 mm and 0.168 mm, respectively. However, the compressive strength of cracked specimens after seawater curing for 56 days is about 30% lower than that of tap water curing specimens. Therefore, although corrosive ions promote the healing of cracks, they could cause significant losses to the mechanical properties of specimens. Maes et al. [72] studied the effect of fracture healing on chloride ion erosion in a seawater environment. Unfortunately, only Wu et al. [4,55] have done relevant research on the healing of CA specimens in seawater. Since the CA is a self-made material, this part is described in Section 3.

River water contains high concentrations of Cl^−^, and deicing salt environment contains ultrahigh concentrations of Cl^−^. In recent years, some researchers [27,62,70,71,72,73] have studied the healing properties of cement-based materials under Cl^−^ environment. For example, Xue et al. [39] found that Cl^−^ first consumed AFm and 3CaO·Al_2_O_3_·CaCO_3_·12H_2_O (the reaction product of AFm and CO_2_ and Ca(OH)_2_) to generate Friedel’s salt, CO_3_^2−^, and OH^−^. This reaction not only promoted the formation of CaCO_3_, but also increased the pH value of the crack solution. The concentration of CO_2_ in the solution also increased, further promoting the carbonization and expansion of the crystal. With the gradual consumption of OH^−^, the pH value of the solution in the fracture gradually decreased. Friedel’s salt will decompose into hydrogarnet at low pH, and hydrogarnet is also sensitive to pH. Hydrogarnet will be carbonized to CaCO_3_ and Al(OH)_3_ at lower pH. The higher Cl^−^ concentration in the healing environment, the more intense the above reaction, the more precipitation of Al(OH)_3_, and the faster the crack closure. CA promoted the formation of AFt. AFt reacted with unhydrated C_3_A (tricalcium aluminate, one of the components of cement) to form AFm, which provided the component that would react with Cl^−^.

The cooling water of geothermal power plants contains high concentrations of Na^+^, Cl^−^ and SO_4_^2−^. S. Monte et al. [47] and Cuenca et al. [45] added CA to concrete and evaluated the healing performance of cracked specimens in this kind of environment. The crack width measured by Cuenca et al. [45] in the experiment was generally less than 0.1 mm and was thus small. However, the crack closure rate of the CA specimen in the test was much lower than that found by other researchers [9,36,39,42,45,47]. Therefore, crack healing performance is poor in such environments, even if CA is added.

## 3. Research on Self-Made CA

The compositions of self-made CA in the existing studies are shown in Table 4. However, only a few studies introduced the reaction mechanism and role of specific components in CA, prefabricating CA, and preventing CA from reacting in advance. Especially for the last one, Lee et al. [74] coated mineral additives with PVA films to prevent an early reaction. Most researchers only introduced the healing properties and healing products of CA. Therefore, in the current research, researchers are expected to have a solid foundation of chemical knowledge.

Park et al. [38] used CaSO_4_ and Na_2_SO_4_ as CA. It was found that because Na_2_SO_4_ has high solubility, the healing products contained ettringite and monosulfate. In contrast, the low solubility of CaSO_4_ led to the absence of sulfate ions in the healing products.

Wu et al. [55] used NaAlO_2_ as CA. Combined with the self-healing mechanism of microcapsules, CA was pressed into blocks and then filled into concrete. This method can effectively avoid the early reaction of CA in concrete mixing and pouring. Subsequently, they studied the healing performance of CA cement paste in seawater. The healing effect was significant, given the crack closure rate of the specimen reached 80% after 1 day of healing. The healing products on the fracture surface were mainly brucite. Since the dissolution and diffusion of A1^3+^ were much slower than those of other ions, the crystalline products (Friedel’s salt, ettringite, and hydrotalcite) of A1^3+^ reacting with Mg^2+^, SO_4_^2−^, and Cl^−^ were mostly accumulated in the CA block. However, compressing CA into a block and filling it into the matrix inevitably impacts the mechanical properties of the matrix. This also increases the implementation difficulty of concrete vibrating and pouring, resulting in the capsule healing agent being difficult to apply in engineering practice.

Li et al. [8] used sodium carbonate, sodium silicate, sodium aluminate, EDTA tetrasodium, and glycine as the masterbatch of CA. In their study, the mechanism of each component was relatively precise. Sodium silicate is soluble in water, and soluble silicate ions react with calcium ions to form a C-S-H gel. Sodium carbonate can react with acidic substances to delay the decrease in the alkalinity of fracture solution. It can also react with calcium hydroxide (Equation (2)), and the generated sodium hydroxide can maintain the alkalinity of concrete and accelerate the solidification of healing products. Sodium aluminate reacts with calcium oxide and gypsum to form ettringite, sodium hydroxide, and aluminum hydroxide (Equations (3)–(5)). EDTA tetrasodium as a calcium chelating agent reduces the interface energy of calcium carbonate crystal. Calcium carbonate will grow into coarse strips with a high aspect ratio which can block cracks more effectively. Glycine is a steel corrosion inhibitor. It was found that the crystal phases of the healing products of the CA specimen and the reference specimen were subject to the same curing condition (underwater curing) with no new mineral species. However, the healing products of the CA specimen contained more C-S-H gel and CaCO_3_ precipitation, while the reference specimen contained more portlandite. This shows that the reaction of Na_2_CO_3_, which is the primary reaction of healing, is rapid.
Na_2_CO_3_ + Ca(OH)_2_ → CaCO_3_ + 2NaOH(2)
Na_2_CO_3_ + Ca(OH)_2_ → CaCO_3_ + 2NaOH(3)
2NaAlO_2_ + 3CaO + 7H_2_O → 3CaO·Al_2_O_3_·6H_2_O + 2NaOH(4)
3CaO·Al_2_O_3_·6H_2_O + 3(CaSO_4_·2H_2_O) + 20H_2_O → 3CaO·Al_2_O_3_·3CaSO_4_·32H_2_O(5)

Park et al. [56] used sodium sulfate and aluminum sulfate as CA. Through the TGA test, it was found that the healing product of the CA specimen contains a large amount of calcite. BSE-EDS analysis of the fracture section showed that CaCO_3_ mainly grows at the crack mouth, and the healing products inside the crack are mainly C-A-H and a small amount of C-S-H and AFm.

Zha et al. [6] used maleic anhydride, sodium hydroxide, and hydrogen peroxide solution to configure CA. After 28 days of curing, almost no healing products could be seen in the crack mouth of the reference specimen, while the CA specimen could completely heal, and the healing products at the crack mouth were mainly CaCO_3_.

## 4. Self-Healing Test

### 4.1. Self-Healing Rate

Currently, the three main aspects of crack healing of cement-based composites are the closure rate of crack mouth, the recovery of mechanical properties and durability of specimens, and the composition analysis of healing products. Table 5 lists the healing ability tests in laboratory experiments. The research on healing efficiency tests can be found in [23,57,75,76].

There are many problems in the observation of crack closure rate by microscope or crack observation instrument. Roig-Flores et al. [49,51] considered that the geometric closure rate of the crack mouth is less reliable than the seepage flow test for the evaluation of the healing effect. Xue et al. [39] found that the calculation of closure rate through the average width of the crack mouth might have a significant error, which further affects the evaluation accuracy of the healing performance. This can be explained as follows: Firstly, the complete healing of a crack’s mouth does not represent the overall healing of crack, as shown by the distribution of healing products in Section 2.3.1. Secondly, due to the heterogeneity of concrete and the variability of crack width along the length direction and depth direction, the test results are too discrete to allow drawing a clear and reliable conclusion. Specific test contents are shown in Table 6. Although there is no consistent conclusion, they have inspired researchers for the future study of CA. Consequently, other researchers could make appropriate improvements to the test method.

Some previous studies [36,39,41,42,45,47,53,77] reported that the CA can improve the mechanical properties and durability of the cracked specimens to a certain extent compared with the reference specimen. However, it is challenging to compare the healing performance of CA due to the differences in the definition of healing rate, the brand of CA, the healing environment, the healing performance test, the water–binder ratio, the cement composition, and other materials added in the experiments. The working conditions of complete healing of cracks or complete stop of leakage in the test were sorted out, including specimens with single CA and specimens mixed with CA and other materials (Table 7).

For the recovery of mechanical properties, some important information is as follows: The density of the self-healing product and the bond strength between the healing product and the original fracture surface are the main factors for the recovery of bending strength [41]. Xue et al. [41] studied the interface between the healing product and the original fracture surface. The authors found that the dynamic viscosity between the healing product and the fracture surface is less than 9 mPa·s (22 °C) and the surface tension of the healing product is 0.026 N/m (22 °C). This property enables the healing product to enter the fracture surface through capillary pressure and bond the fracture surface with the healing product (Figure 6). Sisomphon et al. [36] considered that the main component of the healing product is usually weaker than the C-S-H produced by cement hydration, even if the C-S-H in the healing product is formed by the hydration of unhydrated cement on the crack surface. Moreover, compared with the tensile strength of the matrix, the bond strength between the crack surface and the C-S-H in the healing product is likely to be weak. Xue et al. [39] inferred that the bonding degree and elastic modulus of Al(OH)_3_ (see Section 2.3.2, paragraph 3) crystallization were low; thus, the mechanical properties were not likely to be recovered.

For the recovery of the durability of CA specimens, some important information is as follows: Sisomphon et al. [15] inferred that the formation of calcium carbonate crystals at the crack mouth was beneficial to the impermeability of the structure in the short term. However, it hindered the healing of the internal area of cracks, further hindered the recovery of mechanical properties, and was unfavorable for long-term durability. Escoffres et al. [48] applied a load to the steel bar of the test piece. Cracks were consequently generated on the test piece through the strain of the steel bar. When the reference specimen was cured for 20 h under a constant tensile stress load of 270 MPa, the permeability coefficient of the specimen decreased to 5% of the initial permeability coefficient *k*_0_. In comparison, the healing effect of CA specimens can only reach this level when the CA specimens are maintained under a fixed tensile stress load of 240 MPa for 120 h. After the specimen is completely healed and the permeability coefficient is zero, the load is maintained until the permeability coefficient of the specimen reaches *k*_0_ again. The required stress loads for the reference and CA specimens are 95 MPa and 115 MPa, respectively. This result indicates that although the CA specimen healed slowly, the healing product is more conducive to mechanical recovery than the reference specimens.

Some studies reported the relationship between the above-mentioned healing parameters. Ferrara et al. found that the CA specimens had significant stiffness recovery (stiffness recovery index exceeding 20%) when the crack opening closure rate exceeds 70~80%. Cuenca et al. [45] studied the healing effect of CA specimens under geothermal water healing curing and showed a strong correlation between the crack closure rate and the water absorption healing rate. Monte et al. [47] found that the recovery rate of impermeability is greater than the crack healing rate in the experimental value. The author concluded that the penetration could be stopped when the crack has healed to a certain extent, even if the crack is not completely healed. The restoration of flexural rigidity is the healing of cracks and the healing of internal cracks. Li et al. [8] adjusted the proportion of each material to configure the CA with the best healing performance. In the test, the mechanical properties of a specific ratio of CA recovered optimally, but the recovery of secondary impermeability performance was not optimal.

### 4.2. Factors Influencing Healing Performance

#### 4.2.1. Effect of Moisture Content in Healing Curing Environment

Various studies reported the effects of curing conditions (different relative humidities) on the healing process of CA specimens. The healing effects in descending order are water immersion, wet and dry cycles, and air exposure [28,42,47,48,49,51,77]. The reaction of CA requires the participation of water.

However, some studies reported different results. For example, Sisomphon et al. [36] found that wet and dry cycles (wet 12 h–dry 12 h) are the most suitable conditions for healing concrete and restoring its mechanical properties. Their results can be explained as follows: (1) In the drying stage, as the excess water in the crack evaporates, the ion concentration in the residual water in the crack gradually increases. In this case, the number of reactants for the further reaction is highly concentrated, and the amount of water is also sufficient for the solution reaction. (2) During the drying process, the alkalinity of the solution increases, and the concentration of CO_2_ dissolved in the water gradually increases, which promotes the formation of CaCO_3_ precipitation. (3) Excess water molecules can reduce the surface energy of the hydration product and the van der Waals bonding of the hydration product. Liu et al. [70] used seawater immersion and seawater dry and wet cycles (wet 12 h–dry 12 h, wet 1 h–dry 23 h) to heal the cracked specimens. The crack closure rates were found to be 67.5%, 93.95%, and 27.8% after 56 days of curing. Through TGA and XRD tests, it is found that wet and dry cycles (wet 12 h–dry 12 h) promote the carbonization of Ca(OH)_2_ and the growth of brucite. The healing products also contain a certain amount of halite (NaCl). This experimental result is consistent with that of Sisomphon et al. [36]. Unfortunately, the authors failed to provide convincing explanations for the effect of dry and wet cycle time on the healing of cracks.

For air exposure, the healing effects of the CA specimen and the reference specimen are both poor. Roig-Flores et al. [49] found that the cracks of the CA specimens closed by 17%, while the cracks of the reference specimens expanded by 46%. Escoffres et al. [48] found that the healing effect of the CA specimen is similar to that of the reference specimen in air exposure curing conditions. Ferrara et al. [42] concluded that the CA specimen’s healing effect under natural curing was similar to that of the reference specimen immersed in water through the recovery of crack closure, flexural rigidity, and bearing capacity.

#### 4.2.2. Effect of Temperature

Reinhardt et al. [61] studied the effect of temperature (20~80 °C) on crack healing and obtained the reduction in the normalized flow rate as a function of temperature. Figure 7 is the curve for a crack width of 0.05 mm and a pressure gradient of 1 MPa/m. The authors only analyzed the effect of temperature on the viscosity of water and did not analyze the healing reaction further. For specimens mixed with CA, Ferrara et al. [42] studied the effects of the high-temperature cycle (20~50 °C) and low-temperature cycle (5~20 °C) on self-healing of CA specimens and reference specimens. However, the test values are very dispersed, even if the cracks are classified by width. Roig-Flores et al. [51] found that the healing effect of CA specimens at 30 °C is better than that at 15 °C under the condition of water immersion. The test results of reference specimens are relatively more dispersed and cannot be compared. Therefore, the effect of temperature on the crack healing of cement-based materials and specimens mixed with CA needs further investigation.

#### 4.2.3. Effects of CO_3_^2−^ in Aqueous Solution

Sisomphon et al. [36] immersed the cracked CA specimens in tap water and boiled water for healing maintenance to study the effect of CO_3_^2−^ in the curing water on self-healing. The healing results of the specimens after 28 days of healing maintenance are shown in Figure 8. It can be seen from the diagram that although the concentration of CaCO_3_ in tap water is minimal (120~240 ppm), a large number of CaCO_3_ precipitates appear at the crack mouth of the specimen. At the same time, there is only a small number of healing products at the crack mouth of the specimen cured by boiled water. By testing the bending strength, bending stiffness, and deflection, it was found that the mechanical recovery of the tap water curing specimen is lower than that of the boiled water curing specimen. Although CaCO_3_ precipitation at the crack mouth can improve the permeability resistance of the specimen, it could hinder the further hydration of the incompletely hydrated particles inside the crack. Through SEM-EDS analysis of the healing products inside the crack, it was found that the tap water curing specimens contained more ettringite and the structure of healing products was loose. The boiled water curing specimens contained more C-S-H gel, and the healing products were relatively dense.

Sisomphon et al. [36] also studied the effect of curing water (tap water) replacement (water was replaced once every 12 h) on the self-healing of CA specimens. Regular replacement of curing water made the CO_3_^2−^ concentration relatively constant in the curing environment. It was found that the replacement of curing water slightly improved the mechanical properties of the specimen. Jiang et al. [50] found that the permeability recovery rate and crack closure rate of CA specimens without replacement of water were about twice those of CA specimens with replacement of water after 7 days of healing, while the healing rates of the reference specimens in these two healing environments were almost the same. Sisomphon et al. [15] found that the first replacement of healing water (water replaced once every 7 days) significantly impacted the healing of cracks. The possible reason could be that some healing products dissolve or decompose into the external water environment when the water is replaced in the early stage of healing.

#### 4.2.4. Effects of Crack Characteristics

##### Shrinkage Cracks and Structural Cracks

The cracks can be divided into shrinkage cracks (early age) and structural cracks (service phase) according to the occurrence time. As per some studies [7,38,50,77], long-term curing time increases the hydration degree of specimens and weakens the healing performance of CA. Park et al. [38] inferred that with the increase in curing time, the strength of the specimen increases gradually, and the concentration of Ca^2+^ in the pores stabilizes, but the diffusion rate of Ca^2+^ in pore solution decreases.

However, Li et al. [46] found that the healing rate of shrinkage cracks is less than that of structural cracks. Regardless of shrinkage cracks or structural cracks, the healing effect in the CA specimens is better than that in the reference specimens. Yang et al. [78] and Kan et al. [79] studied the effect of crack age on the self-healing of fiber-reinforced concrete and found a similar situation. It is speculated that more cracks with smaller widths are formed after the mature specimen is loaded; therefore, the specimen heals easily.

##### Several Cracking–Healing Cycles

Structures often simultaneously bear multiple loads during service, which brings significant challenges to concrete healing. By performing tests of several cracking–healing cycles, Cuenca et al. [28] found that the longer the healing time after the first cracking, the better the healing effect after the second cracking. After multiple (5~7) cracking–healing cycles with a period of 1 year, it was found that the cement particles on the crack surface of the reference specimen had almost entirely hydrated, and the specimen could not continue to heal. However, the CA specimens could still maintain high healing properties, and the cracks could still be fully healed. Monte et al. [47] found that the crack closure rate could reach 65~93% after the first cracking–curing cycle. However, after the second cracking–curing cycle, the crack closure rate was only below 20%, even if the specimens were soaked in water for three months. Xue et al. [41] found that the healing product can constitute the heterogeneity of the cement matrix, causing stress concentration and leading to unstable crack propagation. For the reference specimens, recracking between the healing product and the original crack surface occurs more easily. The healed crack can hardly heal completely after recracking, and the flexural strength recovers even less. For CA specimens, if the cracks are well healed, new cracks may be generated after reloading. When the new cracks healed, the flexural strength of the structure increased by nearly 3 times. Sisomphon et al. [36] found that the CA specimens had new cracks during the second cracking.

## 5. Synergetic Effect of CA with Other Components

### 5.1. Synergetic Effect of CA with Fibers

Fiber-reinforced concrete overcomes the shortcomings of brittleness and low strain of ordinary concrete, and fibers can control the width and propagation of cracks to a certain extent. Many researchers have studied the healing performance of CA fiber-reinforced concrete (Table 8). However, most researchers add fibers to the CA specimens to control the crack width of the specimen more efficiently, and the cylinder specimens do not break completely when splitting tensile cracking. Few researchers studied the synergetic effect of CA with fibers or the promoting effect of fibers. Ferrara et al. [44] thought that although the fibers were not directly involved in the chemical reaction of healing, the crystalline healing product formed a combined structure with the fibers. The fibers increase the adhesion point of crystal healing products, promote the bonding of healing products on both sides of cracks, and improve the overall mechanical properties of healing products. The wrapping of healing products protects fibers, especially steel fibers. The rapid crystallization of CA can delay or stop the corrosion of steel fibers.

Cuenca et al. [28] studied the effect of steel fiber arrangement direction on the healing of CA specimens. It was found that the average crack closure rate (77.83%) of specimens with parallel fibers was significantly higher than that of specimens with vertical fibers (54.70%). Guzlena et al. [40] added acrylic polymers and glass fibers to produce specimens with higher bending strength to prevent the specimen from damage. It was found that the film formed by acrylic polymers could prevent the crystal from contacting water. Feng et al. [80] concluded that calcite precipitation could be seen on the PVA fibers inside the cracks due to the influence of polar groups and cell structure of PVA fibers on the nucleation of calcium carbonate.

### 5.2. Synergetic Effect of CA with Expansive Agent

Adding an expansive agent is one of the methods of dealing with the early shrinkage of concrete. According to the chemical composition, the expansive agents can be divided into calcium oxide expansive agent, MgO expansive additive (MEA), and calcium sulfoaluminate based expansive additive (CSA). Among them, the expansion rate of calcium oxide expansive agent is too fast, and the expansion efficiency cannot be effectively exerted. CSA is widely used at present, but MEA is still in the trial stage [81,82,83]. In recent years, some researchers have studied the self-healing effect of MEA on cement-based materials [84,85,86], and others investigated the synergetic healing properties of CA and expansive agents. Sisomphon et al. [15] found that when specimens were only mixed with CSA, the structure of the healing product was poor, although the expansion of the healing product was faster. When specimens were only mixed with CA, the healing process was slow. However, the specimens mixed with CA and CSA showed faster and more effective healing. Table 9 presents the case of specimens mixed with CA and expansive agent.

#### 5.2.1. Crystal Phase of Healing Products

Through the SEM-EDS test, Sisomphon et al. [36] found that the healing products of specimens mixed with CA and CSA were mainly CaCO_3_, C-S-H, and ettringite. By conducting SEM-EDS and XRD tests, Wang et al. [10] found that the healing products of the reference specimens were mainly calcite and aragonite, without other crystals. CaCO_3_, ettringite, C-S-H, and C-A-S-H were healing products of specimens mixed with CA, Na_2_CO_3_, CSA, and CaHPO_4_. The authors reported that ettringite was the hydration product of CA, and Na_2_CO_3_ provided CO_3_^2−^ for the solution in the cracks, which promoted the formation of CaCO_3_ precipitation. Xue et al. [39] studied the synergetic healing performance of CA with MEA, and the reaction mechanism is shown in Figure 9. The authors found that the hydration of CA could promote the dissolution of Ca^2+^ and SiO_3_^2−^ on the crack surface, which would then react with other ions to form gels. MgO would hydrate to produce Mg(OH)_2_. CO_2_ dissolved in water would promote the reaction between Mg^2+^ and CO_3_^2−^ to produce hydrated magnesium carbonate (HMC—MgCO_3_·3H_2_O, 4MgCO_3_·Mg(OH)_2_·4H_2_O, 4MgCO_3_·Mg(OH)_2_·5H_2_O). Compared with CaCO_3_ precipitation, HMC has higher compactness and a denser interconnected microstructure, so it can better contact the surface on both sides of a crack [87,88]. Through the SEM-EDS test, Park et al. [38] found that Al^3+^, SO_4_^2−^, and SiO_3_^2−^ provided by CA and CSA promoted the formation of C-S-H gel and ettringite, while C-S-H gel was not observed in the healing products of reference specimens.

#### 5.2.2. Distribution of Healing Products

Wang et al. [9] split the specimens to analyze the distribution of healing products along the crack depth by SEM-EDS. In the reference specimen, calcite mainly accumulated at the crack mouth. This observation is consistent with the analysis results from Sisomphon et al. [15] detailed in Section 2.3.1. A small amount of calcite was deposited at a depth of 2 mm. The healing products decreased significantly at a depth of 4 mm, and some crystals were non-calcite crystals. When the depth was 6 mm, the healing product was rare, and the crystal morphology of the healing product changed into aragonite. When the depth was 10 mm, there was almost no crystal product. For the specimens mixed with CA, Na_2_CO_3_, CSA, and CaHPO_4_, a considerable amount of calcite grew at the crack mouth, but the crystal form was more regular, and the crystallinity was better. Many crystallization precipitations were attached to the crack surface at the depths of 2, 4, and 6 mm. A small amount of calcite could still be observed at depths of 8 and 10 mm.

Static water was used in the curing environment for the above tests, and some researchers have studied the healing of penetrating cracks under water scouring. Park et al. [38] studied the healing performance of specimens mixed with CA, CSA, and basic magnesium carbonate. It was found that the width of the crack mouth was almost constant after curing healing for 56 days through the seepage test, while all specimens completely stopped leakage (Figure 10). Good healing occurred inside the crack, completely different from the healing described in the previous paragraph.

#### 5.2.3. Healing Performance

Sisomphon et al. [36] found that the deflection deformation of specimens mixed with CA and CSA was about 1.5 times that of CA specimens when the specimens were cured for 28 days. Sisomphon et al. [15] found that only specimens mixed with CA and CSA could heal entirely within 28 days when cracks were 0.3~0.4 mm. Moreover, when the healing time was seven days, the crack closure rate of the specimen mixed with CA and CSA was 3 times that of the CA specimen. Park et al. [38] found that the specimens mixed with CA, CSA, and basic magnesium carbonate (MgCO_3_·Mg(OH)_2_·5H_2_O) could completely heal the cracks with a width of 0.295 mm after five days of healing, and the seepage stopped completely.

### 5.3. Synergetic Effect of CA with Superabsorbent Polymers

SAP has a robust physical swelling capacity. In deionized water, 1 g of SAP can absorb more than 500 mL of water. The absorption rate of SAP is about 10~20 mL/g in the high-alkalinity pore solution of concrete [89,90]. SAP is often used in medical and health fields (e.g., in infant diapers and medical ice bags). In concrete research, SAP is mainly used to alleviate autogenous shrinkage of cement with a low water–binder ratio [91,92,93,94,95]. In recent years, some researchers have studied the healing ability of SAP on cracks [26,89,96,97].

Hong et al. [96,98] found that the rapid swelling of SAP can heal cracks within 5 min and proposed a model to quantify the healing ability of SAP. Li et al. [7] found that the seepage flow rate of the SAP specimen with crack width of 0.4~0.6 mm decreased to 1~2% of the initial flow rate after 15 days of seepage healing. However, the seepage flow rate no longer changed, indicating that the healing stopped. The authors considered that the particle size of SAP after the expansion is large, and the particle size distribution is poor; hence, the seepage path cannot be blocked entirely. When SAP is added, for specimens with crack width less than 0.5 mm, the crack width does not affect the healing efficiency. This observation is entirely different from other test results due to the strong water swelling ability of SAP.

When the surrounding humidity gradually decreased, the expanded SAP colloid gradually released water. Although this can promote the growth of CaCO_3_ crystals on the crack surface, there are two defects. On the one hand, this will make the cement-based matrix have more pores, decreasing the strength of the structure. Hong et al. [96] found that the 28-day compressive strength of specimens with 0.5% and 1% SAP was 79% and 56% of that of the reference specimens, respectively. On the other hand, the healing cracks are likely to crack again under the dry and wet cycles.

Given the above situation, some researchers identified the defects of SAP and improved its performance by adding CA. In the microscopic experiment, Park et al. [38] found that many needle-like products grew in the pit (Figure 11) with the slow release of water in SAP through SEM test. Through optical microscope observation, Li et al. [7] found that the shrinkage of SAP causes a small number of cement slurry fragments and crystal fragments to break from the original structure. These fragments would then block the cracks. In macroscopic experiments, Li et al. [7] found that specimens mixed with Na_2_CO_3_ (CA) and SAP could completely stop leakage after six days of healing. However, when specimens were transferred to dry and wet cycle healing environment, specimens begin to leak again (Figure 12). Furthermore, after two dry and wet cycles, specimens failed to heal completely. Park et al. [38] found that the cracks of specimens mixed with CA and SAP could be completely filled at the beginning of the permeability test. However, with the progress of the experiment, the seepage velocity increased and fluctuated.

When CA and SAP are mixed, different CAs also bring different healing effects. Li et al. [7] selected a variety of CAs to mix with SAP, and the healing effect of the specimen mixed with citric acid and SAP was the best. The cracks with a width of 2 mm could be completely healed after healing for five days. The authors found that citrate ions could be used as the carrier of Ca^2+^ in cement matrix to promote the precipitation of calcium carbonate crystals. However, compared with other specimens, the upper surface of the specimen mixed with citric acid and SAP showed more bulges (Figure 13), and the surface of some specimens even showed spalling. It is considered that the density of SAP particles is relatively low, and SAP particles easily float to the surface of the cement matrix after vibration during pouring. Moreover, the retarding effect of citric acid leads to the expansion of SAP particles.

In summary, SAP particles have a robust, rapid water-stopping effect. It is a new material that all fracture water leakage control researchers dream of. However, the expanded SAP gel was not stable. Although CA can improve the stability, it can be seen from the above experiments that such compensation is still insufficient. It is considered that SAP is not suitable as an admixture to heal cracks, and SAP can be considered a coating to be used on the upstream or downstream face of the structure. When the structure cracks, SAP can quickly close the crack mouth and stop the leakage of the crack. The water released from SAP can also promote the healing of the internal cracks by CA.

## 6. Conclusions

This paper focuses on the autonomous healing of cement-based composite material mixed with crystalline admixtures (CAs). Some conclusions can be drawn as follows:(1)CA promotes the growth of ettringite on the crack surface. The ettringite can heal the cracks and form a network structure that can better cover the crystallization of CaCO_3_. Compared with ordinary specimens, CA specimens show better healing along the crack depth.(2)The growth of CaCO_3_ at the crack mouth is not conducive to the internal healing of cracks and is not conducive to the recovery of mechanical properties of the structure.(3)According to the promoting effect of water content in the curing environment on the healing performance of CA, the order is as follows: appropriate dry–wet cycles (for example, wet 12 h–dry 12 h), water immersion, dry–wet cycles, standard curing, and air exposure.(4)CA showed an excellent synergetic effect with fiber and expansive agent.(5)SAP has a solid ability to repair cracks, but specimens mixed with SAP cannot heal completely. Some researchers found that the addition of CA can make up for this defect to some extent.(6)In the test results, the best healing specimens are as follows: specimen mixed with CA, CSA, and basic magnesium carbonate; specimen mixed with NaAlO_2_; and specimen mixed with Na_2_CO_3_ and SAP.

## 7. Problems and Prospects

The research of CA cement-based composites still needs enormous efforts to tackle the following problems:(1)Currently, there are no relevant standards and norms for evaluating healing performance, causing the calculation method of self-healing rate and healing performance testing to vary greatly among researchers. Consequently, it is challenging to make a horizontal comparison between the research results from different studies.(2)It is not reliable to evaluate the healing effect by crack closure rate. The healing products may gather in the crack mouth or may only gather in the crack interior. Some healing products such as Al(OH)_3_ and Mg(OH)_2_ have poor mechanical properties, while other healing products such as aragonite and hydrated magnesium carbonate have good mechanical properties.(3)For the recovery of the durability of cracked CA specimens, the healing degree is mainly evaluated according to the decrease in seepage flow or seepage pressure ratio. No researchers have studied the effect of CA healing cracks on the infiltration of water and corrosive ions into the concrete through cracks, and no researchers have studied the effect of CA healing cracks on reducing chloride ion corrosion of reinforcement.(4)The composition of CA, healing environment, and other chemical additives determine the crystal phase of healing products, and the crystal phase of healing products could further affect the healing performance. The high confidentiality and variability of the components of commercial CA and the incompatibility of the research results of researchers increase the difficulty of CA research. It is suggested that for commercial CA, the macroscopic healing properties (the recovery of mechanical properties and durability) of CA cement-based composites should be focused on to promote the application of engineering practice better. For the development and research of CA, SEM-EDS, XRD, FTIR, TGA, and other tests should be carried out to analyze the healing products to clarify the single-factor and multifactor coupling effects of each component on the healing products. Then, the macroscopic healing properties of CA cement-based composites should be tested to promote the development and research of CA.(5)The composition and crystal phase of healing products determine the macro healing performance of the structure. There are few studies on the relationship between them in the current research, and there is a lack of systematic research.(6)The cracks produced by the three-point bending, four-point bending, and splitting tensile cracking test are close to the real cracks and the actual working conditions. However, a high degree of dispersion of the test results can occur easily, and reliable conclusions are hard to make. It is suggested that if the test results mainly serve the actual project, the generation of cracks mainly depends on the above methods. If the test is to study some rules of self-healing of CA specimens, more regular cracks can be produced by the method of prefabricated cracks with steel inserts, and the cracks produced by force can be compared and analyzed.(7)There are few studies on the distribution and crystal phase of the healing products of CA specimens along the crack depth, and the healing inside the crack is crucial to the healing of the whole structure. Therefore, the healing behaviors inside the fracture should be further studied, and a CA with faster healing inside the crack than at the crack mouth should be developed.(8)The healing reaction of CA requires the participation of water. This material is highly suitable for underwater buildings, such as underwater shield tunnels, underwater suspension tunnels, and underwater pile foundations. Simulation of underwater conditions should be added.(9)The distribution of healing products through cracks differs, which should be studied further, especially under high water pressure.(10)In some conditions, the study of the healing of cracks needs to consider the bearing condition of the structure, especially for dynamic loads, and cracks are not likely to heal completely.(11)SAP has the characteristics of rapid expansion when exposed to water and shrinkage when the water content around SAP decreases gradually. Although SAP can quickly heal cracks, the compressive strength of specimens that are mixed with SAP was generally decreased. Subsequent studies can focus on the following points: (1) SAP particles have a specific particle gradation, and the expanded SAP gel will be denser. (2) SAP can be used as a coating, and CA can be used as the admixture of concrete. Then, the healing capacity of this kind of specimen should be compared with the specimens mixed with CA and SAP.(12)After cracking, water slowly infiltrates into the cement on both sides of the crack. The expansion of the matrix will also promote the healing of the crack to a certain extent, especially for cement-based materials mixed with CA and expansive agents.(13)Some researchers studied crack healing by numerical simulation. Jiang et al. [99] used CEMHYD3D to simulate crack healing and the effect of crack healing on chloride ion erosion. Xue et al. [100] used XFEM and CS techniques to study the interface between healing products and crack surfaces numerically. Luzio et al. [101] proposed a numerical model for the gradual recovery of mechanical properties of cementitious materials with crack healing. If the numerical simulation results have good compatibility with the experimental results of the healing performance of CA, the numerical simulation can significantly reduce the workload of the experiment.(14)There are few studies on the healing performance of CA in some particular healing environments. Subsequent studies can increase the simulation of working conditions such as seawater, sewage pool, salt fog, freeze–thaw cycle, microbial growth environment, and post-fire.

## Figures and Tables

**Figure 1 materials-15-00440-f001:**
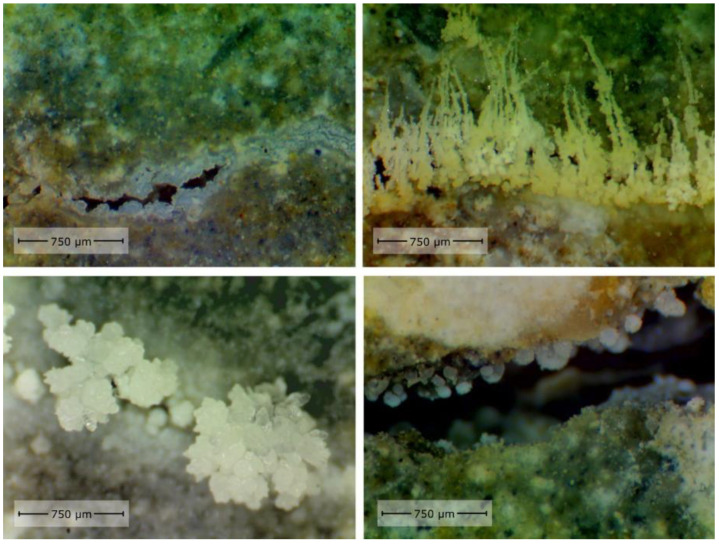
Crystal morphology of healing products. Reprint with the permission from ref. [11]. Copyright 2021 Elsevier.

**Figure 2 materials-15-00440-f002:**
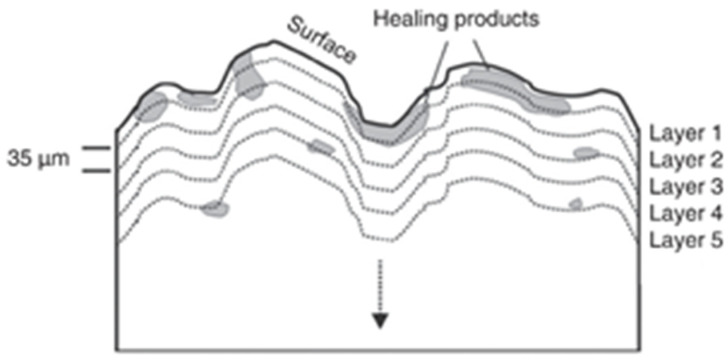
Distribution of healing products along fracture depth direction. Reprint with the permission from ref. [14]. Copyright 2021 Elsevier.

**Figure 3 materials-15-00440-f003:**
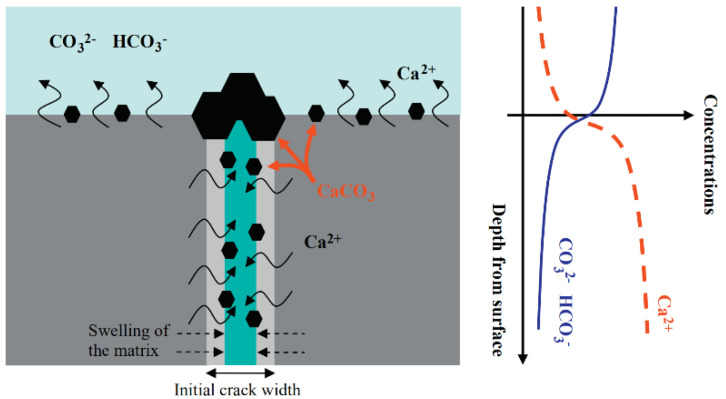
Schematic diagram of the crystallization of calcium carbonate. Reprint with the permission from ref. [15]. Copyright 2022 Elsevier.

**Figure 4 materials-15-00440-f004:**
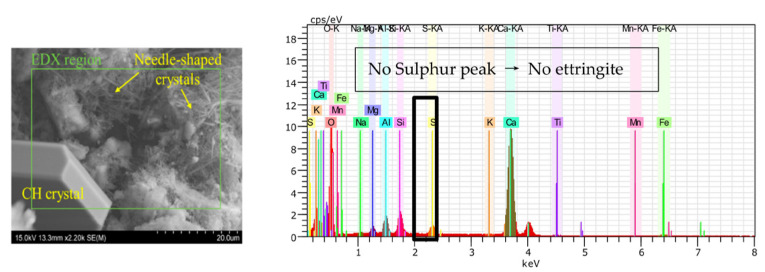
SEM-EDS analysis of needle-like products [64].

**Figure 5 materials-15-00440-f005:**
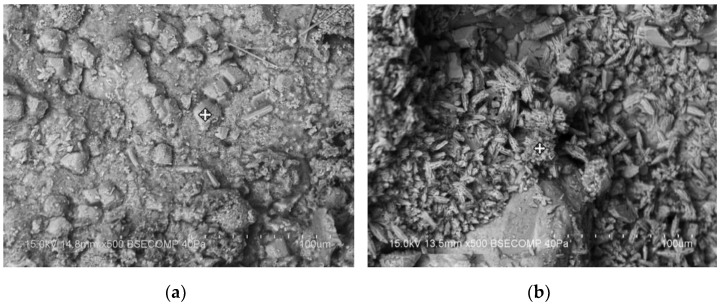
SEM observation of (**a**) calcite and (**b**) aragonite [48].

**Figure 6 materials-15-00440-f006:**
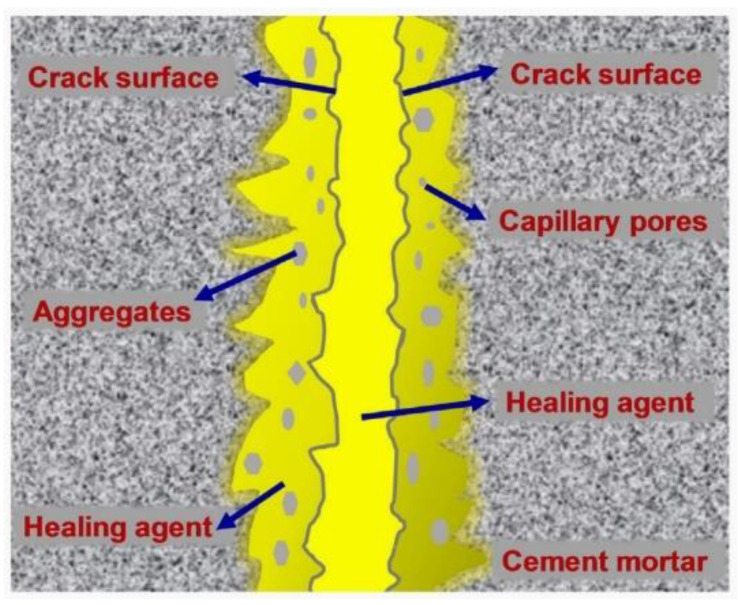
The interface between the healing product and the original fracture surface. Reprint with the permission from ref. [41]. Copyright 2022 Elsevier.

**Figure 7 materials-15-00440-f007:**
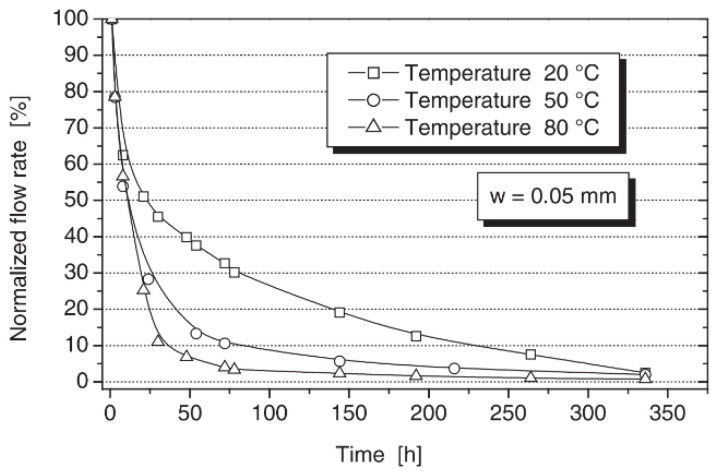
Decrease in the normalized flow rate because of the self-healing of the crack at various temperatures, a pressure gradient of 1 MPa/m, and a crack width of 0.05 mm. Reprint with the permission from ref. [61]. Copyright 2022 Elsevier.

**Figure 8 materials-15-00440-f008:**
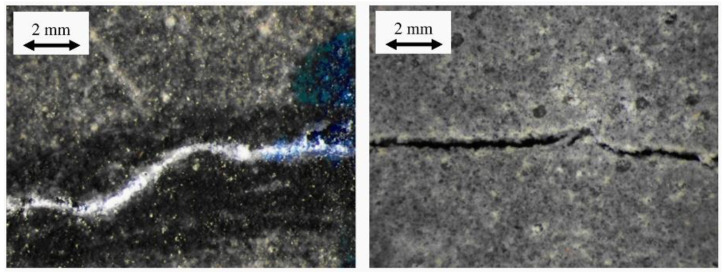
Crack healing results (left: tap water, right: boiled water). Reprint with the permission from ref. [36]. Copyright 2022 Elsevier.

**Figure 9 materials-15-00440-f009:**
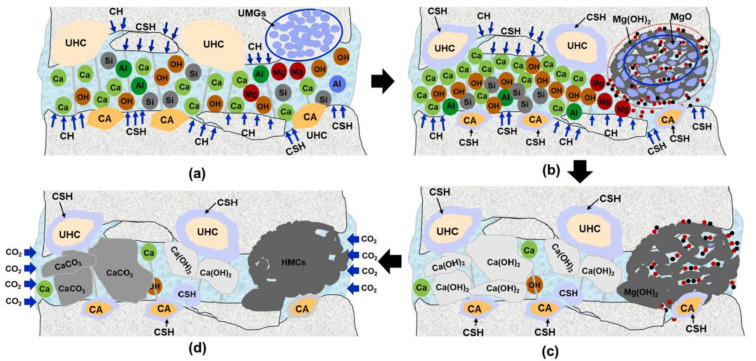
Self-healing mechanism of CA-MEA cementitious material (**a**) the crack has just opened, unhydrated grains exposed to crack solution, (**b**) various ions diffusion, (**c**) crystals formation, and (**d**) carbonation of crystals. Reprint with the permission from ref. [39]. Copyright 2022 Elsevier.

**Figure 10 materials-15-00440-f010:**
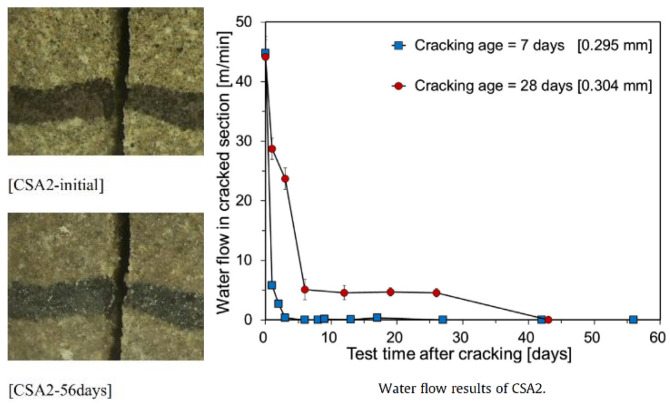
Crack width and water flow of specimens mixed with CA, CSA, and basic magnesium carbonate. Reprint with the permission from ref. [38]. Copyright 2022 Elsevier.

**Figure 11 materials-15-00440-f011:**
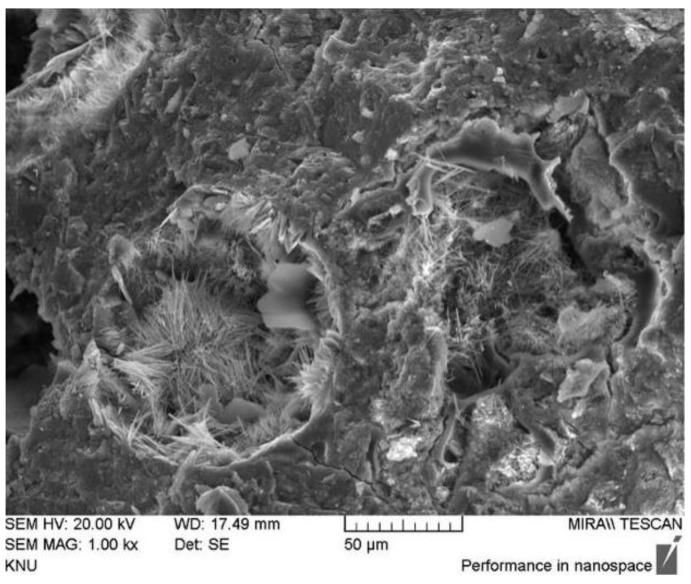
Needle-like products in pits. Reprint with the permission from ref. [38]. Copyright 2022 Elsevier.

**Figure 12 materials-15-00440-f012:**
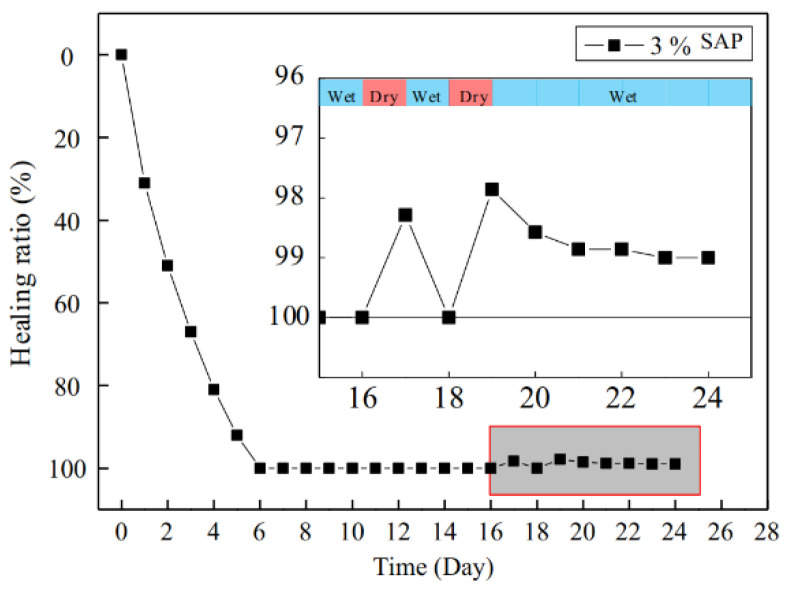
Healing rate of specimens under dry and wet cycle curing. Reprint with the permission from ref. [7]. Copyright 2022 Elsevier.

**Figure 13 materials-15-00440-f013:**
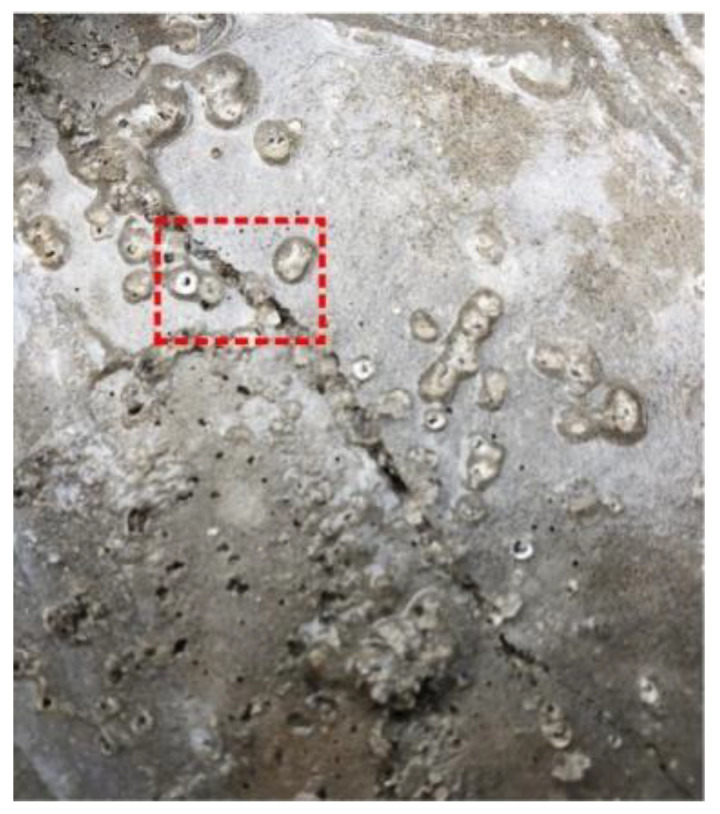
Surface of specimen mixed with citric acid and SAP. Reprint with the permission from ref. [7]. Copyright 2022 Elsevier.

**Table 3 materials-15-00440-t003:** Healing conditions.

Healing Conditions	References
Permeation (water leaking through cracks)	[7,15,31,38,56]
Water immersion	[5,6,9,10,28,29,36,40,42,44,46,48,49,50,51,54]
Wet–dry cycles	[28,36,39,51,52]
Standard curing	[6,44,58]
Climate chamber/humidity chamber	[42,45,49]
Water contact	[49,56]
Air exposure	[28,31,36,42,44,48,49]
Steam curing (80 °C)	[41]
Water immersion (synthetic sea water)	[4,55]
Geothermal water immersion	[45,47]
Wet–dry cycles (0.545 mol/L and 2 mol/L chloride solution)	[39]
Wet–dry cycles (geothermal water)	[45,47]

**Table 4 materials-15-00440-t004:** The compositions of self-made CA.

Self-Made CA	References
CaSO_4_	[38,54]
Na_2_SO_4_	[38,54,56]
NaAlO_2_	[8,55]
Al_2_(SO_4_)_3_	[56]
SiO_2_ (silica fume), citric acid	[7]
Na_2_O·nSiO_2_ (sodium silicate)	[7,8]
Na_2_CO_3_	[7,8,9,10,54]
Maleic anhydride/deionized water/sodium hydroxide solution (concentration of 0.10 mol/L at 90~95 °C)/hydrogen peroxide solution (volume concentration of 30%) = 1:1:1:0.3	[6]
Fumaric acid + Na_2_CO_3_	[29]
NaOH	[5]
Al_2_(SO_4_)_3_, NaHCO_3_, Li_2_CO_3_	[54]
Na_2_CO_3_ + Na_2_O·nSiO_2_ (sodium silicate) + NaAlO_2_ + tetrasodium EDTA + glycine	[8]

**Table 5 materials-15-00440-t005:** Healing ability tests.

Mechanical Tests		Durability Tests		Analysis of Healing Products		Others	
Compressive strength	[6,41,46,58]	Second permeation	[58]	SEM/ESEM	[5,6,7,9,10,28,29,36,38,39,40,41,42,45,46,48,50,54,58]	Crack closure	[4,5,6,7,9,10,15,28,29,31,36,38,39,40,41,45,46,47,49,50,51,52,55]
Tensile strength	[58]	Permeation	[7,15,31,38,47,49,50,51,56]	EDS	[5,6,9,28,36,39,41,42,45,46,48,50,54,55,56]	pH test	[15]
3-point bending	[39,41,42,44,48]	Chloride diffusion	[52]	BSE	[41,56]	ICP	[5,15]
4-point bending	[36,44,47]	Water absorption	[5,29,46,50]	XRD	[4,6,9,10,39,46,50,55]	UPV	[42,44]
Tensile-permeability test	[48]	Gas permeability	[10]	FTIR	[4,39]	MIP	[45]
		Conductivity	[5,15]	TGA	[4,39,41,45,56]	CT	[5]

Note: BSE: backscattered electron; CT: X-ray computed tomography; EDS: energy-dispersive X-ray spectroscopy; FTIR: Fourier transform infrared spectroscopy; ICP: inductively coupled plasma; MIP: mercury intrusion porosimetry; SEM/ESEM: (environmental) scanning electron microscopy; TGA: thermogravimetric analysis; UPV: ultrasonic pulse velocity; XRD: X-ray diffraction.

**Table 6 materials-15-00440-t006:** Large discreteness of test results.

Test Contents	References
Initial crack width, healing ratio	[52]
Crack closure rate	[10]
Crack closure rate	[45]
Crack geometry (average width, maximum width, area, and closure rate of cracks), water seepage	[49,51]
Crack width, crack closure rate, stiffness recovery rate	[47]
Crack width (average width, maximum width, minimum width), initial water seepage	[56]
Crack width, crack closure rate, curing condition, healing period, number of crack-healing cycles	[28]

**Table 7 materials-15-00440-t007:** Complete healing of cracks or complete cessation of leakage.

Components	Crack Width	Healing Conditions	Healing Age	Reference
1.5% CA + 10% CSA	0.3~0.4 mm	Water immersion	28 days	[15]
1% CA	0.1 mm	Water immersion	30 days	[39]
0.5% CA (self-made)	0.32 mm	Standard curing	28 days	[6]
2% Na_2_CO_3_ + 3% SAP	0.2 mm	Permeation	4 days	[7]
5% Na_2_SO_4_	0.23 mm	Permeation	4~21 days	[56]
1.5% CA + 10% CSA + 2.5% basic magnesium carbonate	0.295 mm	Permeation	3 days	[38]
1% CA + 10% MEA	0.1 mm	Water immersion (0.545 mol/L Cl^−^ solution)	40 days	[39]
1% CA	0.1 mm	Water immersion (2 mol/L Cl^−^ solution)	30 days	[39]
5% CaO-NaAlO_2_	0.39~0.44 mm	Seawater	7 days	[55]
0.8% CA + 0.025% cellulose nanocrystals/cellulose nanofibrils	0.1 mm	Geothermal water	3~6 months	[45]

Note: CSA: calcium sulfoaluminate based expansive additive; MEA: MgO expansive additive; SAP: superabsorbent polymer.

**Table 8 materials-15-00440-t008:** The case of specimens mixed with CA and fibers.

Fibers	References
Steel fibers	[28,42,44,47,48,49,51]
Polyvinyl alcohol (PVA) fibers	[5,36,39]
Glass fibers	[40]
Amorphous fibers	[47]
Cellulose nanocrystals, cellulose nanofibrils	[45]
Alumina nanofibers	[45,53]

**Table 9 materials-15-00440-t009:** Case of specimens mixed with CA and expansive agent.

Expansive Agent	References
CSA	[9,10,15,36,38,54,56]
MEA	[39]

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
