# Peer review of "Influence of Crystalline Admixtures and Their Synergetic Combinations with Other Constituents on Autonomous Healing in Cracked Concrete—A Review"

_materials, 2022, doi:10.3390/ma15020440_

Round 1

Reviewer 1 Report

The manuscript is a state-of-the-art review document on the influence of crystalline admixtures on the self-repair properties of cracked concrete. In this sense, it  systematizes knowledge in this scientific area, identifies knowledge gaps, and suggests possibilities for future research in this domain.   Bearing in mind that this aims at being a review document, originality consists in the systematization of information, seeking to condense the main advances in this domain in order to facilitate the work of future researchers.   The review is sufficiently detailed to be of interest for researchers within this field of expertise. However, there are several recently published previous review works on the use of crystalline additions to promote self-repair of concrete. Some of these documents are not cited in the present manuscript, and it would make sense to include references to the following ones:   Ravitheja, A., Reddy, T.C.S. & Sashidhar, C. Self-Healing Concrete with Crystalline Admixture— A Review. J. Wuhan Univ. Technol.-Mat. Sci. Edit. 34, 1143–1154 (2019).
https://doi.org/10.1007/s11595-019-2171-2   Mohd Nasim, U.K. Dewangan, Shirish V. Deo, Autonomous healing in concrete by crystalline admixture: A review, Materials Today: Proceedings, Volume 32, Part 4, 2020, Pages 638-644, ISSN 2214-7853, https://doi.org/10.1016/j.matpr.2020.03.116.
  Nele De Belie, Elke Gruyaert, Abir Al-Tabbaa, Paola Antonaci, Cornelia Baera, Diana Bajare, Aveline Darquennes, Robert Davies, Liberato Ferrara, Tony Jefferson, Chrysoula Litina, Bojan Miljevic, Anna Otlewska, Jonjaua Ranogajec, Marta Roig-Flores, Kevin Paine, Pawel Lukowski, Pedro Serna, Jean-Marc Tulliani, Snezana Vucetic, Jianyun Wang, and Henk M. Jonkers, A Review of Self-Healing Concrete for Damage Management of Structures. Adv. Mater. Interfaces 2018, 5, 1800074, https://doi.org/10.1002/admi.201800074.   Among the suggestions for possible future research, I highlight the current nonexistence of standards for evaluating healing performance that allow objective comparison of different products performance. Another interesting remark is that the effect crystalline admixtures healing cracks on reducing chloride ion corrosion of reinforcement is not yet quantified. These two recommendations (among some others), based on the review work, represent innovative information worth publishing.
  The paper is well organized and well-written, containing only a couple of typos and format issues that may be checked and corrected.    

Author Response

The following have been added to the manuscript by reading these articles:

  1. The main influencial factors of crack healing include: 1) the composition of cementitious materials, water binder ratio and age of matrix; 2) healing environment; 3)width and shape of cracks.
  2. Some studies have noticed that the mechanical strength of CA specimens are 7%~10% higher than that of ordinary specimens.
  3. Various manufacturers have provided a series of crystal material products, mainly including: surface-applied spraying or coating, integral waterproofing admixtures and sealing and repairing mortars. Engineers can choose appropriate crystalline materials according to the specific circumstances of the building.

Reviewer 2 Report

The paper is an extensive review that shows influence of crystalline admixtures and their synergetic combinations with other constituents on autonomous healing in cracked concrete.
The issue has been studied in many research centers.
The authors of this paper made a very careful review of the literature and the research carried out. 
Then they showed the results of their own research - the results of research of self-made CA and self-healing test. The composition, healing mechanism, and crystallization products of commercial crystallization admixture are shown.
In my opinion, the most interesting is to show the synergetic effect of CA with other components - with fibers, expansive agent and super absorbent polymers.
Conclusions (points 1 - 6) are of great practical importance, mainly due to the increase in durability of the structure.
The problems and prospects shown at the end give hope for more interesting papers on this subject.
I consider the paper valuable and recommend that you publish it in mdpi journal materials with minor revision included.

Minor revision:
The number of cited references is much larger than what is needed to describe the topic. I propose reducing references to the most representative ones.

Author Response

Dear reviewer,

Thank you very much for reviewing my article.

Some references with little relevance to the topic have been deleted.

Reviewer 3 Report

Self-healing of concrete is a promising research path to increase the durability of concrete structures. Several ways are being investigated to increase the natural healing potential of concrete. The present manuscript exposes a literature review about ‘Influence of crystalline admixtures (CA) and their synergetic combinations with other constituents on autonomous healing in cracked concrete’

Thi literature review will probably be of interest to the journal readers and the self-healing community. The manuscript is mostly clearly structured in seven parts: an introduction giving some background material, a second part concerning commercial crystallization admixture, a third part about research of self-made CA, a fourth part about self-healing tests, a fifth part dedicated to synergetic effect of CA with other components, a sixth part which is the conclusion, and a seven part which is a part exposing some problems and prospects.

Overall, the manuscript is well structured and clear. Among other minor comments, the authors should address to main issues to increase the readability of their work. First, a clear distinction should be made between studies dedicated to self-healing using CA and other research works about autogenous self-healing. Then, some material about autogenous healing should be added or moved to the introduction section. Especially, the authors should talk about the use of supplementary cementitious materials, which can increase the healing potential (please find references below), and move the discussion about the location of the healing products (lines 140 to 162 in the manuscript).

List of the comments the authors should address:
- l 33 – 34: please add a reference supporting the hydration depth of cement particles. Precise the w/c ratio
- l 37: CSH can also grow in the crack and participate in the self-healing process. SEM images are reported in the literatture
- between lines 52 and 53: as the authors report experimental studies with calcined clay aggregates, it is recommended to add a small paragraph about the current use of SCMs to increase the healing potential (especially slag and ternary binders)
10.1016/j.conbuildmat.2012.07.026 (also give maximum healable crack widths)
10.1016/j.conbuildmat.2016.03.087
10.1016/j.jobe.2021.102739
- Table 1: please revise table caption to mention ‘commercial CA’. Is it possible to add price ranges for these CA and their typical wt% in concrete?
- Table 2: precise in the caption these results are from XRF
- l 99: please explain what M and R stand for
- l 140 – 162: this discussion about the location of the healing products in autogenous healing does not seem to be correctly placed, in my opinion, as it is hard to find related CA results. Maybe it would be better to recap these findings in the introduction
- l 184: if calcium carbonate, and more specifically calcite, is the main hydration product, this paragraph should be moved first in the section
- Table 4: Can metakaolin-based aggregates be considered as CA?
- l 347: please do not refer to mechanical regains as they are not reported in Table 7
- l 381-393: please rearrange the paragraph to make a clear distinction between impermeability and mechanical regains
- l440: please precise Reinhardt result is related to autogenous healing
- l 620: maybe replace ‘heal’ by ‘fill’ because the healing result is really quick
- l 691: about SAP the authors state that ‘its shrinkage caused by water loss is a great defect for cement-based materials’. It is advised not to write such a strong statement. A deeper explanation related to the general decrease of compressive strength associated with SAP would be more beneficial
- l 763: the part about mechanical regains associated with healing is unclear. As it is far from the focus of the article, it is suggested to remove this point.
